# CD276 as a Candidate Target for Immunotherapy in Medullary Thyroid Cancer

**DOI:** 10.3390/ijms241210019

**Published:** 2023-06-12

**Authors:** Kinga Hińcza-Nowak, Artur Kowalik, Agnieszka Walczyk, Iwona Pałyga, Danuta Gąsior-Perczak, Agnieszka Płusa, Janusz Kopczyński, Magdalena Chrapek, Stanisław Góźdź, Aldona Kowalska

**Affiliations:** 1Department of Molecular Diagnostics, Holycross Cancer Centre, 25-734 Kielce, Poland; arturko@onkol.kielce.pl; 2Endocrinology Clinic, Holycross Cancer Centre, 25-734 Kielce, Poland; a.walczyk@post.pl (A.W.); iwonapa@tlen.pl (I.P.); danutagp@o2.pl (D.G.-P.); aldonako@onkol.kielce.pl (A.K.); 3Division of Medical Biology, Institute of Biology, Jan Kochanowski University, 25-406 Kielce, Poland; 4Collegium Medicum, Jan Kochanowski University, 25-319 Kielce, Poland; stanislawgo@onkol.kielce.pl; 5Surgical Pathology, Holycross Cancer Centre, 25-734 Kielce, Poland; agnieszkapl@onkol.kielce.pl (A.P.); januszko@onkol.kielce.pl (J.K.); 6Faculty of Natural Sciences, Jan Kochanowski University, 25-406 Kielce, Poland; magdalena.chrapek@ujk.edu.pl; 7Clinical Oncology, Holycross Cancer Centre, 25-734 Kielce, Poland

**Keywords:** CD276, immunotherapy, medullary thyroid cancer

## Abstract

Medullary thyroid cancer (MTC) is a rare malignancy, and the treatment of metastatic MTC is challenging. In previous work, immune profiling (RNA-Seq) of MTC identified CD276 as a potential target for immunotherapy. CD276 expression was 3-fold higher in MTC cells than in normal tissues. Paraffin blocks from patients with MTC were analyzed by immunohistochemistry to confirm the results of RNA-Seq. Serial sections were incubated with anti-CD276 antibody, and scored according to staining intensity and the percentage of immunoreactive cells. The results showed that CD276 expression was higher in MTC tissues than in controls. A lower percentage of immunoreactive cells correlated with the absence of lateral node metastasis, lower levels of calcitonin after surgery, no additional treatments, and remission. There were statistically significant associations of intensity of immunostaining and percentage of CD276 immunoreactive cells with clinical factors and the course of the disease. These results suggest that targeting this immune checkpoint molecule CD276 could be a promising strategy for the treatment of MTC.

## 1. Introduction

Medullary thyroid cancer (MTC) is the third most common thyroid cancer, accounting for 3–5% of all thyroid cancer cases [1]. MTC originates from the parafollicular cells (C cells), which produce calcitonin (CT) and several other peptides. Among these is a carcinoembryonic antigen that is used as a non-specific tumor marker for the surveillance of MTC patients. CT level is a diagnostic and prognostic factor, and an indicator of remission in MTC [2]. MTC can occur sporadically (75% of cases) or in a hereditary form (25% of cases) as part of the multiple endocrine neoplasia (MEN) type 2 syndromes MEN2a and MEN2b. Surgery plays a major role in the treatment of MTC. Total thyroidectomy is performed in all patients, and cervical LN dissection is performed depending on serological, imaging, and intra-operative findings [3]. Metastasis to central and latero-cervical lymph nodes occurs in up to 90% of patients with tumors > 4 cm in diameter [4,5]. Distant metastases are present at diagnosis in approximately 10% of MTC patients and detected at a higher rate (19–38%) during follow-up [6]. The treatment of patients with locally advanced or metastatic disease remains problematic. The 10-year overall survival (OS) rate in unselected MTC patients is approximately 75%, whereas it is 40% in patients with locally advanced or metastatic disease. Patients with locally advanced disease are treated with external beam radiation therapy (EBRT), whereas those with metastatic disease are treated with tyrosine kinase inhibitors (TKIs), which increase progression-free survival but do not improve OS [7,8,9]. The highly selective inhibitor of the receptor tyrosine kinase RET, selpercatinib, is more effective than other TKIs because it specifically targets the mutated RET protein, thereby reducing the risk of side effects [10]. However, resistance to approved drugs has been reported. Although the use of immunotherapy for MTC has been investigated, there are no approved drugs to date [7,8,9].

Identifying new treatments with minimal side effects is important. In recent work from our group, immune profiling of MTC tumors identified CD276 as a potential target for immunotherapy [11]. CD276 is an immune checkpoint molecule that functions as a T cell inhibitor, promoting the immune escape of tumor cells by, among other things, reducing the secretion of tumor necrosis factor alpha, interferon gamma, and other cytokines. Increasing evidence suggests that CD276 has synergistic effects with other immune checkpoints. The use of combination therapy (anti-CD276 antibody and anti-PD1/PD-L1 antibody) for cancer treatment may be promising [12]. This study is a case series, in which we investigated the expression of CD276 in MTC and the correlation of immunohistochemistry (IHC) staining intensity with histo-clinical features.

## 2. Results

### 2.1. Characteristics at Presentation and Primary Treatment

Patient demographics and the clinicopathological features of the 46 cases are presented in Table 1. The study group were comprised of 31 women (67.4%, 31/46) and 15 men (32.6%, 15/46). The median age at diagnosis was 52 years (range, 24–84). The median tumor size was 14 mm (range, 2–100). Most patients (69.6%, 32/46) did not have tumor multifocality. Gross type extrathyroidal extension was detected in 6.5% of patients (3/46). Angioinvasion was present in 10.9% (5/46) of patients. Lymph node metastases and distant metastases were present in 51.3% (19/46) and 4.3% (2/46) of patients, respectively.

### 2.2. Response to Therapy and Final Follow-Up

An excellent response to primary therapy was observed in 24 patients (52.2%). Twenty patients (43.5%) showed a biochemical incomplete response to initial therapy, whereas two patients (4.3%) showed a structural incomplete response. Twelve patients (26.1%) required additional therapy during the course of the disease. In addition to the EBRT in 12 patients, four required surgical re-treatment (8.7%). One of the 12 patients required additional use of selpercatinib (2.2%). There were four deaths among cases with MTC, but only two cases were MTC-related (Table 2). The Kaplan–Meier curve of overall survival in the years following surgery are shown in Figure 1. The 5-, 10- and 15-year survival rates for the entire cohort were 93.37%, 93.37% and 83%, respectively.

### 2.3. Relationship between the Intensity of Immunostaining and the Percentage of CD276 Immunoreactive Cells and Histopathologic Factors

Statistical analysis of the IHC results was performed by dividing the study group into two subgroups. The first subgroup included those with CD276-positive samples with either 1+ or 2+ intensity and 11–50% immunoreactive cells, whereas the second group included those with samples with 2+ or 3+ intensity and 51–100% immunoreactive cells. Thirteen samples (28.3%) had a staining intensity of 1+. Intensity of immunostaining 2+ and 3+ was observed in 24 (52.2%) and 9 (19.6%) samples, respectively. The number of patients and the percentage of CD276-positive cells were as follows: 20 (43.5%) for 11–50%, 21 (45.7%) for 51–75%, and 5 (10.9%) for 76–100% (Table 3). No CD276 immunoreactive cells were observed in the tissues outside the tumor (Figure 2A–C).

The association between the intensity of immunostaining and the percentage of immunoreactive cells with CD276 expression and histopathologic factors of MTC is presented in Appendix A. Samples were subjected to univariate analysis; multivariate analysis was not possible because of the size of the study group. Univariate analysis suggested statistically significant correlations between higher intensity of immunostaining (2+ or 3+) and higher percentage of immunoreactive cells for CD276 (51–100%) and individual factors such as gender, lateral lymph node metastasis, and postoperative serum CT levels. Increased expression of CD276 was observed more frequently in male patients (*p* = 0.0255). Lateral lymph node metastasis (*p* = 0.0010) was more frequent in patients with 2+ or 3+ intensity of immunostaining and 51–100% of CD276 immunoreactive cells. The number of patients with increased CD276 expression postoperatively who had serum CT levels ≥ 2 was 3 times higher than the number of other patients (*p* = 0.0036). More than 2 times more germline mutations in the RET gene occurred among patients with 2+ or 3+ intensity of immunostaining and 51–100% of CD276 immunoreactive cells than in others (*p* = 0.02321). A less than excellent response to initial therapy was observed in 22 patients, and this was significantly associated with 2+ or 3+ intensity of immunostaining and 51–100% of CD276 immunoreactive cells (*p* = 0.0374). Furthermore, statistical analysis showed that fewer patients from the 51–100% CD276-positive group achieved disease remission than other patients (*p* = 0.0053), and these patients were more likely to require additional therapies (*p* = 0.0293). The order of variables in terms of their impact was assessed based on their odds ratio (OR), calculated in relation to outcome variables at final follow-up (or initial response to therapy). OR values indicate that lateral lymph node metastasis is the most important variable in both clinical contexts (univariable ORs of 11.45 and 7, respectively) (Appendix A). Accordingly, sensitivity, specificity, positive and negative predictive values, and accuracy of the most critical predictors in terms of predicting response to initial therapy were assessed for this variable (Appendix A). No statistically significant correlation was observed with any other features analyzed.

## 3. Discussion

MTC is a rare tumor, accounting for 5% of all thyroid cancers, and its prognosis is worse than that of papillary thyroid cancer. The reported 10-year disease-specific mortality rate for MTC varies from 13.5% to 38% [13,14]. MTC is responsible for 13.4% of all deaths from thyroid cancer [15]. Surgery is the only curative treatment for MTC. The OS rate of patients with distant metastasis at the time of diagnosis is 40% [16]. For patients with distant metastasis, there is no effective therapy or curative option. Chemotherapy has limited response rates of approximately 20% [6]. Vandetanib and cabozantinib are the only multikinase inhibitors (MKIs) approved for the treatment of advanced MTC [17], although no MKIs have been shown to increase OS. New next-generation, small-molecule TKIs designed for highly potent and selective targeting of oncogenic RET alterations were recently developed [18]. An example of a selective drug is selpercatinib, which is a subject of clinical trials. Three patients in the present study group required systemic treatment because of progressive disease caused by distant metastases to the liver, lung, and bone. Two patients received classical chemotherapy because MKIs were not available at the time. Patients died from MTC in 2007 and 2010. One patient is currently being treated with selpercatinib in a clinical trial. In the last few years, immunotherapy has transitioned from a promising to a well-established therapeutic option in several types of malignancies because of its effect as an immune checkpoint inhibitor. 

CD276, also known as B7 homolog 3 (B7-H3), is an immune checkpoint molecule and immunoregulatory protein [19,20]. In 2001, Chapoval et al. first cloned and characterized the cDNA encoding CD276 [21]. Human CD276 is located on chromosome 15q24.1 and encoded by a single 4.1 kb mRNA. The protein exists either as a transmembrane protein or as a soluble isoform. Transmembrane CD276 is a type I transmembrane protein that contains 316 amino acids and has a molecular weight of 45–66 kDa. The transmembrane form is mostly present in the cytoplasm of tumor cells [20,21]. Transmembrane CD276 is found not only on the surface of tumor cells, but also in cytoplasmic vesicles and in the nucleus. Ingebrigsten et al. showed that nuclear CD276 is involved in colon cancer progression and metastasis, suggesting that nuclear CD276 could be a useful prognostic marker in colon cancer [22]. Soluble CD276, which is cleaved from the surface by a matrix metallopeptidase or produced through the alternative splicing of the intron, has also been detected in the serum of cancer patients, suggesting its potential as a noninvasive biomarker [23,24]. CD276 mRNA is expressed widely in normal human tissues, whereas the CD276 protein is rarely present; the difference between the mRNA and protein expression patterns suggests the existence of a specific post-transcriptional regulation mechanism. However, the molecular mechanisms regulating CD276 expression remain unclear [25]. By contrast, the CD276 protein is overexpressed in many types of malignancies (breast cancer, lung cancer, ovarian cancer, brain tumors, gastric cancer, and squamous cell carcinoma) and is correlated with poor prognosis, increased tumor grade and metastasis, drug resistance, recurrence rate, and decreased OS [26]. In this study group, CD276 was expressed in all MTC samples; the number of positive cells was at least 11% and staining intensity was at least 1+. Because of the low number of C cells in the healthy thyroid gland and their diffuse occurrence, CD276 was not detected in normal cells. CD276 was initially characterized as a T cell-stimulating protein, although most current studies describe it as a T cell inhibitor that promotes tumor aggressiveness and proliferation. Thus, CD276 may be an important immunological target in cancer [19,21,25,27]. The role of CD276 in the development of MTC and during its clinical course has not been evaluated to date. In this study, we showed that a higher immunostaining intensity and a greater percentage of CD276 immunoreactive cells were associated with certain clinical factors related to a severe course of the disease, response to therapy, and the status at the end of treatment. Assessment of the relationship between CD276 expression in tumor cells and clinical factors demonstrated its effect on lateral lymph node metastasis and postoperative serum CT levels, underscoring the need to identify effective therapies for patients with MTC and elevated CD276 protein levels. In the present cohort, 12 patients required additional therapy including surgical re-treatment, EBRT, and selpercatinib. CD276 has been widely studied in non-small cell lung cancer (NSCLC). Studies show that CD276 expression is related to invasion, metastasis, proliferation, and the prognosis of NSCLC patients [28]. Yonesaka et al. indicated that 74% of tumor samples express CD276 with a staining pattern of 1+, 2+, or 3+ as assessed by IHC. Anti-CD276 immunotherapy combined with anti-PD-1/PD-L1 antibody therapy is a promising approach for the treatment of CD276-expressing NSCLCs [29]. In the present study, patients who had distant metastases at the time of diagnosis or during the subsequent course of the disease were characterized by a high percentage (51–75%) of CD276 immunoreactive cells and 2+ intensity of immunostaining, suggesting the potential of anti-CD276 therapy. Arigami et al. demonstrated that CD276 expression in primary breast tumors is significantly correlated with increased tumor size and lymphovascular invasion [30]. CD276 was also evaluated in relation to its involvement in the development of ovarian cancer. Zang et al. showed that CD276 is expressed in 93% of ovarian tumors, as determined by IHC staining. In addition, analysis of cumulative survival time and recurrence revealed that samples with CD276-positive tumor vasculature are associated with a significantly shorter survival time and a higher rate of recurrence [31]. Kasten et al. showed that preclinical radioimmunotherapy in ovarian cancer using CD276 antibodies radiolabeled with ^212^Pb α-particles targets tumor cells and the vasculature, thus, showing promising effects along with low toxicity [32]. Li et al. showed that CD276 promotes gastric cancer cell migration and invasion, and its overexpression increases tumor infiltration depth [33]. Wang et al. showed that high CD276 expression is associated with advanced TNM stage and lymph node metastasis in patients with esophageal squamous cell carcinoma [34]. High expression of CD276 in the tumor vasculature may also contribute to the development of metastasis [35]. Expression of vascular endothelial growth factor (VEGF) in tumor cells is mediated by the soluble form of CD276, suggesting that the combination of anti-CD276 and anti-VEGF drugs is a potential therapeutic approach [24]. The high expression of CD276 in tumor tissues has generated interest among researchers. Abundant experimental evidence supports that CD276 affects tumor progression. The differential expression of CD276 in tumors and healthy tissues suggests that targeting CD276 could be an effective strategy. Several clinical trials of CD276 are currently underway. The results of a clinical trial with enoblituzumab (anti-CD276 antibody; MGA271) show that it possesses anticancer properties. MGA271 was well tolerated with no dose-limiting toxicity and no severe immune-related side effects [36]. Clinical studies of combination therapy with anti-CD276 antibodies and chemotherapy or other checkpoint inhibitors may demonstrate that targeting CD276 in cancer is a synergistic treatment approach [37].

In conclusion, the results of this study indicate that CD276 affects the development and course of MTC. Due to the size of the study group and the single-center nature of the study, the results obtained may not be representative of the population, which may lead to a lack of generalizability. However, given the strong correlation between clinical factors and both the intensity of immunostaining and the percentage of CD276 immunoreactive cells, there is a need for further research, which may contribute to the design of effective treatments for patients with MTC. 

## 4. Materials and Methods

### 4.1. Patients

The study included 46 patients with MTC (42 living and four dead patients) who were selected from the thyroid cancer database at the Endocrinology Clinic of Holycross Cancer Centre, Kielce (HCC). The medical records of patients were reviewed to obtain demographic information, the histological and clinical characteristics of the cancer, treatments, responses to treatment, and clinical course. The initial response to therapy was evaluated at 3 months after surgery. Responses were classified as excellent, biochemical incomplete, or structural incomplete [3]. The final follow-up was on 31 December 2021, and the patients were placed in the following categories: no evidence of disease (NED), biochemical incomplete, structural incomplete, death/MTC-related, and death/MTC-unrelated. Archival paraffin blocks of MTC were obtained from the Pathology Department of Holycross Cancer Center, and two pathologists (J.K. and A.P.) verified the MTC diagnosis independently. Paraffin blocks selected by the pathologist were cut into serial sections and processed for IHC. All study procedures were approved by the Institutional Review Board of Jan Kochanowski University, Kielce, Poland (approval number: 12/2020).

### 4.2. Immunohistochemistry

Serial sections of formalin fixed, paraffin-embedded tissue samples from 46 patients with MTC were stained with monoclonal anti-CD276 antibody (Invitrogen, Carlsbad, CA, USA, 6A1). Briefly, staining was performed on the Ventana BenchMark XT (Ventana Medical Systems Inc., Tucson, AZ, USA). The staining protocol included online deparaffinization, HIER (Heat Induced Epitope Retrieval) with Ventana Cell Conditioning 1 for 32 min, and primary antibody incubation for 20 min at 31 °C. Antigen-antibody reactions were visualized using the Ventana OptiViewTM Amplification kit, followed by Ventana OptiViewTM Universal DAB Detection Kit (Optiview HQ Linker, 8 min; Optiview HRP Multimer, 8 min; Optiview Amplifier H_2_O_2_/Amplifier, 4 min; Optiview Amplifier Multimer, 4 min; Optiview H_2_O_2_/DAB, 8 min; Optiview Copper, 4 min). Counterstaining was performed using Ventana Hematoxylin II for 8 min, followed by bluing reagent for 4 min. Finally, slides are removed from the stainer, dehydrated, and coverslipped for microscopic examination. Positive controls included a known CD276 positive human tonsil tissue. IHC slides were scored by two independent pathologists (J.K. and A.P.). The scoring criteria for staining were based on the intensity of immunostaining and the percentage of immunoreactive cells, and the percentage of CD276-positive cells and the intensity of the staining were scored. IHC-CD276 staining expression was scored as “weak” (1+), “moderate” (2+), and “strong” (3+), and samples were divided into three groups according to the percentage of CD276-positive cells as follows: 11–50%, 51–75%, and 76–100%; <10% positive cells was considered a negative result.

### 4.3. Statistical Analysis

Continuous data were expressed as medians, quartiles, and range (minimum and maximum). Categorical data were summarized by frequencies and percentages. Group comparisons were performed using the chi-square test or Fisher’s exact test for categorical variables and the Mann U-test for quantitative variables (due to the lack of normality in the Shapiro–Wilk test).

Statistical tests were two-tailed and a *p*-value < 0.05 was considered significant. All statistical analyses were performed using R (version 4.0.3; The R Foundation for Statistical Computing, Vienna, Austria).


**Study strength**


Our study had several strengths. Given the currently available treatments for advanced and metastatic forms of MTC, which in a great many cases are met with the emergence of resistance among patients, it is very important to search for new treatments that will increase the chances of progression-free survival with a concomitant improvement in patients OS. In addition, new treatments should have the lowest possible rate of side effects, so conducting research in their search should focus on highly selective molecules. The present study addresses the role of CD276 in the development and clinical course of MTC, which has so far been unknown. The results considered here are the first report in the literature on the importance of CD276 in MTC and may contribute to improving treatments for this type of cancer.


**Study limitations**


This study has two main limitations. The first limitation is the size of the study group. MTC is quite a rare disease, as it accounts for less than 5% of all thyroid cancers, which obviously affects the number of patients available to study. However, because this type of cancer is relatively rare, the results of this study offer new and attractive information for this group of patients. The second limitation is related to the single-center nature of the study, which may contribute to the limited generalizability of its results; however, given its novel nature, its clinical value remains high.


**Future direction**


Deeper knowledge is needed to determine the specific impact of CD276 on the development and clinical course of MTC. In addition, there is a need to increase the size of the study group. The present study included patients whose final follow-up was on 31 December 2021. Additional needed is generalize the results by performing analyses based on data from other centers, which involves conducting a multicenter study. This requires obtaining acquisition of samples from other centers and analysis of newly obtained data.


**Practical value**


Extensive literature evidence indicates that CD276 influences tumor progression. Increased expression of CD276 in tumor tissues is of interest to researchers. In turn, the difference in CD276 expression between cancer and healthy tissue suggests that targeting CD276 may be an effective therapeutic strategy. The our data identified an immune checkpoint, which may facilitate the development of new treatments in MTC. Several clinical trials are currently underway focused on CD276, and the results obtained in this study may be the basis for their continuation in this type of cancer.


**Conclusions**


CD276 influences the development and course of MTC. In addition, it is worth noting that CD276 plays a role in the development of many types of cancer, including breast cancer, lung cancer, ovarian cancer, brain tumors, gastric cancer, and squamous cell carcinoma), and its elevated expression correlates with a worse clinical course. Therefore, CD276 can be considered as a versatile candidate for immunotherapy, capable of being used as a molecular target in the treatment of many types of cancer.

## Figures and Tables

**Figure 1 ijms-24-10019-f001:**
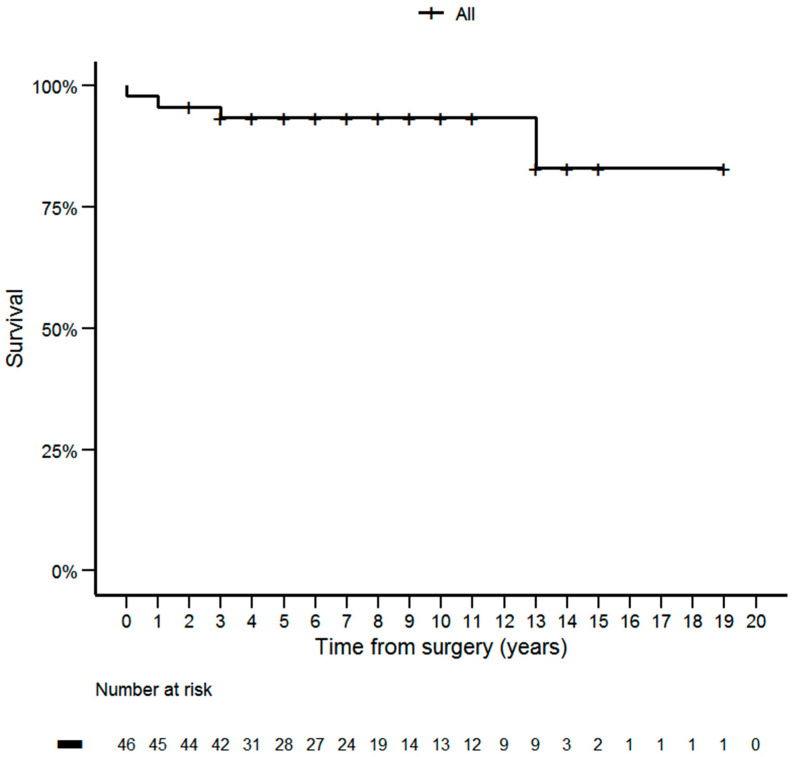
The Kaplan-Meier curve of overall survival in the years following surgery.

**Figure 2 ijms-24-10019-f002:**
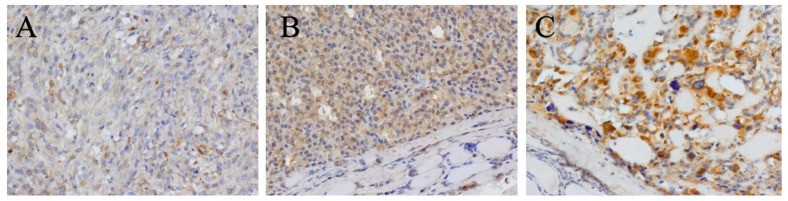
Detection of CD276 by immunohistochemistry in MTC tumors samples. Representative staining intensities of scores 1+ (weak) (**A**), 2+ (moderate) (**B**), and 3+ (strong) (**C**). Original magnification, ×200.

**Table 1 ijms-24-10019-t001:** Characteristics of the study group.

Feature	Total (*n* = 46)
Gender, *n* (%)	
Female	31 (67.4%)
Male	15 (32.6%)
Median age at diagnosis, years (Q1,Q3; range)	52.0 (41.0, 61.8; 24.0–84.0)
Median tumor size, mm (Q1,Q3; range)	14.0 (7.0, 23.0; 2.0–100.0)
Multifocality, *n* (%)	
No	32 (69.6%)
Yes	14 (30.4%)
Extrathyroidal extension, *n* (%)	
No	43 (93.5%)
Yes	3 (6.5%)
Angioinvasion, *n* (%)	
No	41 (89.1%)
Yes	5 (10.9%)
Tumor stage, *n* (%)	
T1a	12 (26.1%)
T1am	8 (17.4%)
T1b	9 (19.6%)
T1bm	3 (6.5%)
T2	6 (13.0%)
T2m	4 (8.7%)
T3	3 (6.5%)
T3m	1 (2.2%)
Node stage, *n* (%)	
N0	27 (58.7%)
N1a	5 (10.9%)
N1b	14 (30.4%)
Distant metastasis, *n* (%)	
M0	44 (95.7%)
M1	2 (4.3%)

T, tumor; N, node; M, metastasis.

**Table 2 ijms-24-10019-t002:** Initial response to therapy and response at follow-up.

Initial Response to Therapy	*n* (%)
Excellent	24 (52.2%)
Biochemical incomplete response	20 (43.5%)
Structural incomplete response	2 (4.3%)
Final Follow-Up	*n* (%)
NED	23 (50.0%)
Biochemical incomplete	17 (37.0%)
Structural incomplete	2 (4.3%)
Death MTC-unrelated	2 (4.3%)
Death MTC-related	2 (4.3%)
Additional Therapies	*n* (%)
No	34 (73.9%)
Yes	12 (26.1%)

NED, no evidence of disease; MTC, medullary thyroid cancer.

**Table 3 ijms-24-10019-t003:** Intensity of immunostaining and the percentage of immunoreactive cells with CD276 expression.

	IHC-CD276 (1+ or 2+) (11–50% Cells)	IHC-CD276 (2+ or 3+) (51–100% Cells)	Total (*n* = 46)
	*n* = 20	*n* = 26	*n* = 46
Intensity of IHC-CD276 expression			
1+	13 (65.0%)	0 (0.0%)	13 (28.3%)
2+	7 (35.0%)	17 (65.4%)	24 (52.2%)
3+	0 (0.0%)	9 (34.6%)	9 (19.6%)
Percentage of CD276-positive stained cells			
11–50%	20 (100.0%)	0 (0.0%)	20 (43.5%)
51–75%	0 (0.0%)	21 (80.8%)	21 (45.7%)
76–100%	0 (0.0%)	5 (19.2%)	5 (10.9%)
Intensity of immunostaining and the percentage of immunoreactive cells			
1+ (11–50%)	13 (65.0%)	0 (0.0%)	13 (28.3%)
2+ (11–50%)	7 (35.0%)	0 (0.0%)	7 (15.2%)
2+ (51–75%)	0 (0.0%)	17 (65.4%)	17 (37.0%)
3+ (51–75%)	0 (0.0%)	4 (15.4%)	4 (8.7%)
3+ (76–100%)	0 (0.0%)	5 (19.2%)	5 (10.9%)

IHC, immunohistochemistry.

## Data Availability

Data are available on request due to restrictions privacy and ethical.

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
