# Peer review of "CD276 as a Candidate Target for Immunotherapy in Medullary Thyroid Cancer"

_ijms, 2023, doi:10.3390/ijms241210019_

Round 1
Reviewer 1 Report
1- Please provide study design. Is this case series manuscript or random or non-random clinical trial.
2-Please provide appropriate strobe statement and strobe checklist as supplementary file.
3-Please provide following headings such as study strength, study limitation, future direction, study implication, practical value and conclusion. All heading must be written in detail with explanation.
4-It seems a non-intervention study right? Please explain in details on your study design and illustrate the flow diagram.
5-Please provide two heading such as instrumentation and second is chemical and provide the list of items used in study along with the source.
6-The current content in the introduction is too little. Please provide theoretical background of study.
okay
Author Response
Question/Comment 1
Please provide study design. Is this case series manuscript or random or non-random clinical trial.
Thank you reviewer for this remark.
This is case series study, which is mentioned in the Introduction (page 2).
Question/Comment 2
Please provide appropriate strobe statement and strobe checklist as supplementary file.
Thank you reviewer for this remark.
Added as supplementary file.
Question/Comment 3
Please provide following headings such as study strength, study limitation, future direction, study implication, practical value and conclusion. All heading must be written in detail with explanation.
Thank you reviewer for this remark.
Supplementing in manuscript (pages 8-9).
Question/Comment 4
It seems a non-intervention study right? Please explain in details on your study design and illustrate the flow diagram.
Thank you reviewer for this remark.
The study was restrospective. It was not an intervention study. All the details of the study have already been included in the Materials and Methods section. Flow diagram added as supplementary file.
Question/Comment 5
Please provide two heading such as instrumentation and second is chemical and provide the list of items used in study along with the source.
Thank you reviewer for this remark.
In the Materials and Methods section described all of items used in study along with the source. This includes the reagents and equipment used to perform the IHC procedure (Page 8; section: 4.2. Immunohistochemistry).
Question/Comment 6
The current content in the introduction is too little. Please provide theoretical background of study.
Thank you reviewer for this remark.
The introduction clearly outlines why new treatments should be sought in patients with advanced MTC. The main problem in the treatments available so far is the emerging resistance to available drugs, and there is a need to look for new solutions for patients. I attach the relevant excerpt from the introduction: Patients with locally advanced disease are treated with external beam radiation therapy (EBRT), whereas those with metastatic disease are treated with tyrosine kinase inhibitors (TKIs), which increase progression-free survival but do not improve OS [7–9]. The highly selective inhibitor of the receptor tyrosine kinase RET, selpercatinib, is more effective than other TKIs because it specifically targets the mutated RET protein, thereby reducing the risk of side effects[10]. However, resistance to approved drugs has been reported. Although the use of immunotherapy for MTC has been investigated, there are no approved drugs to date [7–9].
Identifying new treatments with minimal side effects is important. In recent work from our group, immune profiling of MTC tumors identified CD276 as a potential target for immunotherapy.

Reviewer 2 Report
This work shows how CD276 immunoreactivity in medullary thyroid carcinoma (MTC) correlates with clinical variables and disease course: there is a statistically significant association between such clinical factors and both the intensity of immunostaining and the percentage of CD276 immunoreactive neoplastic cells. I think this work is well written, supported by statistically significant data and its strength lies in the clinical correlation. However, the authors should also discuss some non-statistically significant results, for example in the sentence “that patients from the 51–100% CD276-positive group less frequently achieved disease remission (p = 0.0053)” the p value is not statistically significant. Morevorer, in the Supplementary Table 3, they should add p values for sensitivity, specificity, positive and negative predictive values and accuracy. Furthermore, in the Abstract section, the authors should clearly explain that RNAseq was used in the previous study that showed a higher CD276 expression in MTC cells than in normal tissues. Lastly, they should discuss the limitations of their work. In fact, the study cohort is small and size cohort limits the scientific value of their results. Also, the single-center nature of the study may limit the generalizability of their results.
I think that this manuscript is well written and needs only minor adjustments, in particular a punctuation and spelling check to eliminate typos
Author Response
Question/Comment 1
The authors should also discuss some non-statistically significant results, for example in the sentence “that patients from the 51–100% CD276-positive group less frequently achieved disease remission (p = 0.0053)” the p value is not statistically significant.
Thank you reviewer for this remark.
The obtained results according to applied test is statistically significant. In addition, they put a sentence at the end of the section (2.3. Relationship between the intensity of immunostaining and the percentage of CD276 immunoreactive cells and histopathologic factors), where it clearly says that there is no correlation with other features, and all the others that correlate are described in the text: No statistically significant correlation was observed with any other features analyze (Page 6).
Question/Comment 2
Morevorer, in the Supplementary Table 3, they should add p values for sensitivity, specificity, positive and negative predictive values and accuracy.
Thank you reviewer for this remark.
Added in Supplementary Table 3.
Question/Comment 3
Furthermore, in the Abstract section, the authors should clearly explain that RNAseq was used in the previous study that showed a higher CD276 expression in MTC cells than in normal tissues.
Thank you reviewer for this remark.
Added as supplementary in the abstract; line 2 of abstract.
Question/Comment 4
Lastly, they should discuss the limitations of their work. In fact, the study cohort is small and size cohort limits the scientific value of their results. Also, the single-center nature of the study may limit the generalizability of their results.
Thank you reviewer for this remark.
Added as supplementary; (Section: 3. Discussion; Page 7).

Round 2
Reviewer 1 Report
The authors did not made sufficient changes as per request. one statement is not a details.
Please write one paragraph minimum on issues that were raised.
The content is incomplete
Author Response
Thank you to the reviewer for all the comments. Regarding review report from second round, we have completed the missing issues. Supplementing in manuscript (pages 8-9).
We hope you will find the revised version acceptable for publication and look forward to hearing from you soon.
Sincerely yours,
Kinga Hińcza-Nowak
Round 3
Reviewer 1 Report
publishable
no